# New Evidence about the Polychromy of Early Imperial Cycle from the *Augusteum* of *Rusellae* (Tuscany)

Paolo Liverani [1,*], Susanna Bracci [2,†], Roberta Iannaccone [3], Sara Lenzi [4] and Donata Magrini [2]

1 Department of History, Archaeology, Geography, Figurative and Performing Arts, University of Florence, 50121 Florence, Italy
2 Institute of Heritage Science, ISPC-CNR, 50121 Florence, Italy
3 Department of Chemical, Physical, Mathematical and Natural Sciences, University of Sassari, 07100 Sassari, Italy
4 Department of Civilisations and Forms of Knowledge, University of Pisa, 56126 Pisa, Italy
* Correspondence: paolo.liverani@unifi.it
† This paper is dedicated to the memory of our beloved colleague and friend Susanna Bracci who passed away during this research.

**Abstract:** This study is concerned with recent analyses of seven marble statues from the imperial cycle of the Augusteum of Rusellae, in the south of Tuscany, Italy. The sculptures represent the deified couple Augustus and Livia, Livilla, Claudius, an anonymous young girl and a headless cuirassed emperor (maybe Domitian). In addition, a fragment of a leg, from another cuirassed statue, was also considered. All of them are preserved in the city of Grosseto, in the Maremma Archaeology and Art Museum. Still preserved traces of polychromy and gilding were investigated both in situ, using non-invasive portable techniques, and in laboratory, taking two micro-samples. The non-invasive approach was based on multi-band imaging techniques (Vis, UVL and VIL) integrated with analyses (XRF, reflectance spectroscopy). A portable optical microscopy was also used for documenting the analysed areas. Two micro-samples from the gilding decoration of the headless cuirassed statue were also analysed using EDS-SEM. Comparing the results from the analytical survey, important information about the use of ochres and Egyptian blue on the cuirassed headless Emperor statue has been highlighted with the presence of gilding in the cuirasses and in the mantle, enriching the knowledge of this important imperial cycle, in addition to contributing to the archaeological point of view.

**Keywords:** sculptural polychromy; Roman portraits; Rusellae; archaeometry; ND techniques; pigments; gilding

## 1. Introduction

### 1.1. Rusellae and the Augusteum

The Etrusco-Roman town of Rusellae lies in Tuscany, near Grosseto, not far from the Tyrrhenian coast. The site was extensively excavated in the second half of the past century when a part of the Etruscan necropolis and the Roman city were discovered.

One of the most important areas brought to light is the *forum*, located in a little valley between two hills. It shows different phases [1]. A sestertius was found in the Republican drainage system covered by the Julio-Claudian paving. Its chronology, 18–15 BC [2] (p. 131, pl. XIV), [3] (p. 108, n. 117), allows us to date the renovation and enlargement of the square to those years. New monumental buildings were added at that time, including the Basilica [4], on the eastern side of the *forum*, and the Augusteum in its south-western corner [5] (pp. 594–599), [6] (p. 677), [7] (pp. 538–540), [8], [9] (pp. 846, 863–864), [10] 415–418, A 74, Figure 27, [11] (pp. 69–76, nn. 20.1–30), [12] (pp. 187–192), [13] (pp. 170–174), [14] (pp. 116–118, cat. 45).

The Augusteum, discovered in 1966, is a hall with an apse on the eastern wall and niches on the sides. The entrance was not from the *forum*, but from the western *portico* of the square (Figure 1).

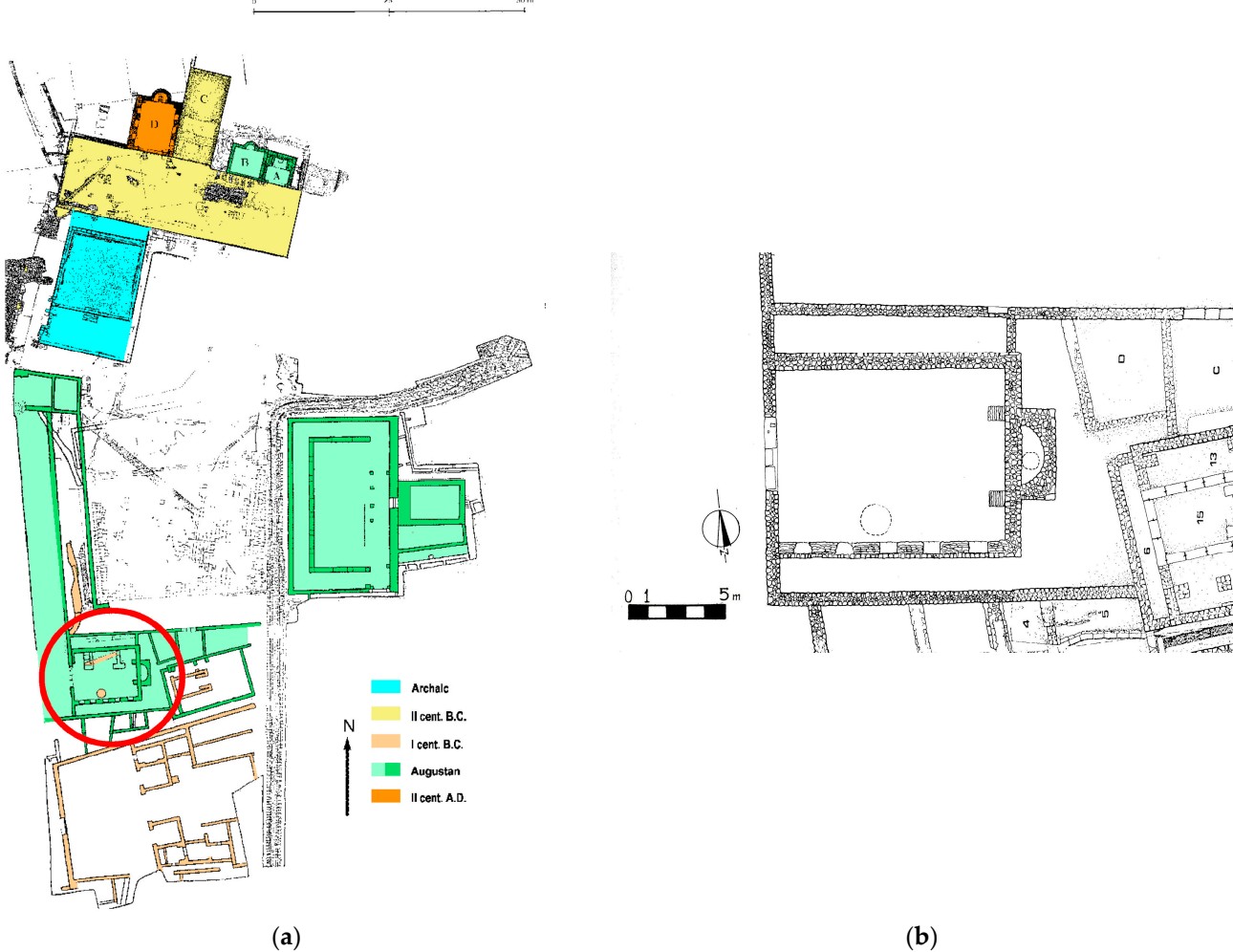

**(a)** **(b)**

**Figure 1.** (**a**) The *forum* of *Rusellae* with the *Augusteum* (author: P. Liverani); (**b**) plan of the *Augusteum* (from Nicosia, F.; Poggesi, G. Eds., Roselle. Guida al parco archeologico, NIE: Siena, 1998).

The hall was decorated with a rich series of sculptures: without considering a head of Helios, found near the entrance, all the others were portraits of the imperial family, spanning from the Julio–Claudian to the Flavian age. A seated statue of a more-than-life-sized divus Augustus as Jupiter (inv. 97737) was originally displayed in the eastern apse [11] (p. 69, n. 20.2). Probably during Claudius' reign, the apse was closed with a wall and a similar more-than-life-sized portrait of the diva Livia in the guise of Caeres (inv. 97772) was set near Augustus against the wall on their brick bases [11] (p. 69, n. 20.1).

Some female portraits [5] (pp. 596–598) (among them, Iulia Livilla, inv. 97740) [11] (p. 70, n. 20.7) and the upper part of a cuirassed statue, tentatively identified with the emperor Domitian [5], (pp. 597–598); [15], (p. 94, n. 15, pp. 97–98); [16], (p. 28, cat. no. IIa 3, pl. 14, 3); [17] (p. 16); [18] (p. 411 note 36); [14] (p. 118); [11] (p. 70 cat. no. 20.12, pl. 60,4); [19]; [20] (pp. 126–127), were found near the northern wall, while some portraits of young boys [5] (p. 598) and Octavia [11] (p. 70, n. 20.11) were discovered near the southern wall. In the central part of the hall, there was a leg of the cuirassed statue. Other fragments such as the portraits of Drusus the elder [11] (p. 69, n. 20.3) and Antonia the younger [11] (p. 69, n. 20.4) and the lower part of the cuirassed sculpture were found in a cistern under the hall, which is probably part of a late-republican house replaced by the *Augusteum*.

Three more portraits, Tiberius [11] (p. 71, n. 20.28); [21] (p. 44, n. 328), Drusus the younger [11] (p. 71, n. 20.30); [21], (p. 62, n. 552), and Agrippina the elder [11] (p. 71, n. 20.29); [21] (p. 71, n. 641), were found immediately to the south of the hall, in the House of the Mosaics [21,22]. It is possible they were originally set in the *Augusteum*.

If we also consider these three sculptures from the House of the Mosaics, we reach the number of twenty imperial portraits originally set in the *Augusteum*. Nowadays, they are preserved in the Maremma Archaeology and Art Museum in Grosseto. Their display tries to respect the original position they probably had inside the *Augusteum*.

Notwithstanding the fragmentary state of part of them, this nucleus is the richest series of this type in Italy.

We have several elements to reconstruct the architectural frame in detail. Among the findings, there is a significant amount of plaster fragments which allow us to understand that the hall had an elaborate decoration. The lower part of the walls was decorated with marble veneer. In the upper part, the plaster was decorated with painted stucco: there remain fragments of a big moulding, probably the cornice which crowned the marble revetment, and minor fragments belonging to the frames and pediments of the niches. Most of them are painted in a few bright colours such as blue, red, yellow, brown [23]. All these elements give us an idea of the background and the setting of the gallery of statues. For this reason, one of the aims of the project is to achieve a 3D reconstruction of the *Augusteum* which considers both the decoration of the walls and the original position of every sculpture, including their colours.

### 1.2. Polychromy

Despite the lack of information about the presence of original colour either soon after the excavation or in later publications, some traces of ancient pigments are still visible with the naked eye, more than half a century after the discovery of the cycle.

The traces of colours were preliminarily documented with a portable microscope. They were preserved in particular on the hair, the robes and other elements of some sculptures: Augustus, Livia, Iulia Livilla, the headless cuirassed emperor and the leg fragment of a second cuirassed sculpture.

### 1.3. Aim of the Project

In the frame of the project for the edition of the *Augusteum*, the whole cycle underwent a series of non-invasive analysis for the identification of the traces of the original polychromy. There are several reasons of interest in the study of the colours of this gallery: first, we have the opportunity of studying the colours not of a single sculpture, but of an entire group. To date, only a few similar cases have existed: one of them is the imperial cycle from Veleia (now in the National Museum of Parma), which has traces of colours, in particular red, black and yellow; however, due to the old age of the excavation (1761), we know very little about the context [24]. Other case studies are groups of sculptures which were found in a similar context, such as the group of portraits from the Room of Fundilia, at the Diana sanctuary at Nemi. These were accurately studied by Amalie Skovmøller and the sculptures show clear traces of colours on robes and hair [25]; [26] (pp. 43–114). Recently studied examples of comparable sculpture groups are the marble decorations from Oplontis, Villa A [27], and the later marble reliefs from the *mithraeum* of the *Castra Peregrinorum* in Rome [28].

## 2. Materials and Methods

To analyse the tiny traces of polychromy, an in situ and totally non-invasive protocol was applied [29]. In addition, two micro-samples have been also taken to study and integrate results obtained using the non-invasive approach. Photographic multiband imaging (MBI) was applied in the first phase to document and collect preliminary data on the spatial distribution of pigments, restoration materials on the surfaces and conservation

status. Based on an already well-established MBI manual [30], visible, ultraviolet-induced visible luminescence (UVL) and visible-induced luminescence (VIL) were acquired.

For the MBI techniques, two cameras were used: a digital Canon EOS 7D (18 Mpixel) (Canon, Tokio, Japan) to acquire visible reflected and UVL images, and for VIL acquisitions, a modified camera Canon EOS 400D (10.10 Mpixel) (Canon, Tokio, Japan) modified through the removal of the internal ICF/AA filter. Both cameras were equipped with a Canon EF-S 18–135 mm f/2.8 IS lens (Canon, Tokio, Japan), and as radiation sources, two Quantum T5D-R flashes (Quantum Instruments, Bartlett, IL, USA) were used, equipped with a QF80 Qflash UV/IR Wave Reflectors with a filter holder (Quantum Instruments, Bartlett, IL, USA). To provide a proper output and input wavelength band for each technique, different combinations of filters were used. Every specific combination of filters applied on flashes and on the lens is summarized in Table 1.

**Table 1.** Summary of the combination of radiation sources and filter(s) used for each of the multispectral imaging techniques considered.

| Technique | Camera | Filters on the Flashes | Filters on Objective |
| --- | --- | --- | --- |
| Ultraviolet-induced visible luminescence (UVL) | Canon EOS 7D | B + W 403 UV black | B + W 486 UV/IR cut |
| Visible reflected (VIS) | Canon EOS 7D | B + W 486 UV/IR cut | B + W 486 UV/IR cut |
| Visible-induced luminescence (VIL) | Canon EOS 400D | B + W 486 UV/IR cut | B + W 093 IR |

To provide calibrated and reproducible images, a Colorchecker (X-Rite ColorChecker Passport Photo, X-Rite, Grand Rapids, MI, USA) and a Spectralon® Diffuse Reflectance target (Labsphere Inc., North Sutton, NH, USA) were used. All the images were acquired in RAW format to preserve the information and to avoid the camera software elaboration. The output images were converted in TIFF format with $2413 \times 3619$ pixels resolution at 8-bit.

In a second phase, addressed by the photographic documentation, some areas were selected and analysed using X-ray fluorescence spectroscopy (XRF) and fibre optics reflectance spectroscopy in the UV–VIS range (FORS) to reconstruct the original paint palette.

The X-ray fluorescence spectra were collected using a hand-held Bruker Tracer III-SD spectrometer (p-XRF) (Bruker, Mannheim, Germany), equipped with Rh anode, SDD detector (FWHM < 145 eV at 100.000 cps). The spectra were acquired for 60 s with the following working parameters: 40 kV, 12 μA. The analysed area is an ellipse of $3 \times 4$ mm. Spectra were analysed with the ARTAX software (v. 7.4.0.0).

Fibre optic reflectance spectroscopy (FORS) spectra in the 350–900 nm range were acquired using a Tungsten source (Ocean Optics mod. HL200, Ocean Insight, Orlando, FL, USA) and a spectrometer (Ocean Optics mod. HR2000, Ocean Insight, Orlando, FL, USA) equipped with optical fibres. The measuring head with 0° illumination and 0° signal collection allowed the acquisition of the reflectance spectrum of an area of about 2 mm$^2$. Each acquired spectrum was the average of 30 scans. As a reference, a Spectralon© plate (WS-1S-L Labsphere certified standard, 99% reflective material, Labsphere Inc., North Sutton, NH, USA) was used. The acquired reflectance spectra, handled with OOIBASE software (v. 32), were compared with references included in both ISPC-CNR spectral database and IFAC-CNR FORS spectral databases of pigments [31,32].

The documentation of the measured areas and the preliminary autoptic observation of the surfaces was performed using a portable optical microscope, mod. Scalar DG-2A with a 25–200× optical zoom. All the photos were acquired at a magnification of 25× (area of $13 \times 8$ mm). In order to integrate the results on the gilding areas on the cuirasses of the headless emperor statue, two micro-samples were also taken to investigate the technique. The samples were observed under the optical microscope; then, the images and the spectra of the two micro-samples were acquired with an ESEM Quanta200 (Environmental Scanning Electron Microscopy) FEI/Philips Electron Optics (formerly FEI, Hillsboro, OR, USA). The measurements were carried out in low vacuum (1 Torr) using

an accelerating voltage of 25 kV and a working distance of 10 mm. Spectra and images (Backscattered and secondary electrons) have been acquired using CIGSEDS software (v 3.00).

## 3. Results

In the group of statues investigated during the campaign, most show little traces of colour, especially red. Traces of red are visible on the statue of Augustus, on the sculpture of Iulia Livilla, Germanicus' daughter, on the hair and the robe of Livia, on the drapery of the young girl, probably Octavia the younger, and on the skin of the head of Claudius. A red colour is also present on a fragment of the leg of the second cuirassed statue.

On the same areas, the ultraviolet luminescence photography (UVL) shows dark areas with high absorbance behaviour for UV radiation, suggesting the presence of an iron-based pigment as a red ochre. FORS and XRF spectra confirm the presence of an iron oxide-based pigment. For a complete description of each statue, please see the Supplementary Materials.

The UVL images on the marble surface of the statues also show an intense yellow luminescence, which mainly corresponds to areas without visible traces of colours. The most likely explanation is the presence of an organic material applied in the past conservative interventions (private communication with the conservator). This material survived in the most degraded and porous areas. The hypothesis of luminescence from a residual ground layer is not supported, due to the presence of this behaviour also on the brown encrustation on the surface and in the fractures. In some areas of the sculpture, the yellow fluorescence is coincident with the deposition of dust and incoherent material related to burial.

In addition to the luminescence of the incoherent deposits, binder used to stick pieces together and some adhesive tape residues are also visible, as in the left side of Claudius' face (Figure 2).

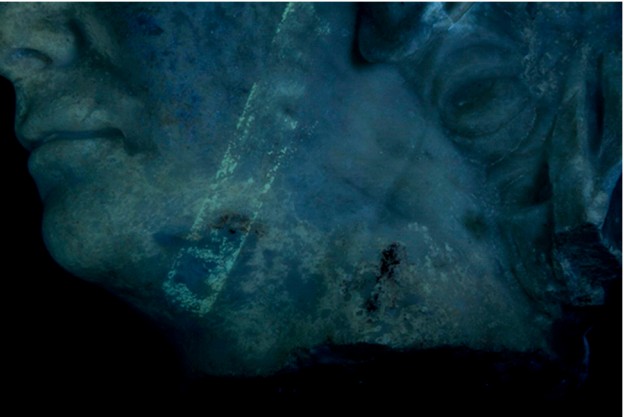 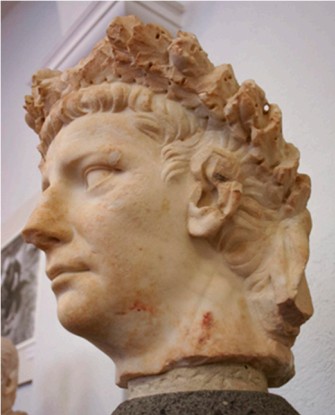

**Figure 2.** UVL detail and visible image of the residual adhesive on the left side of Claudius' face.

The analyses on the statue of cuirassed headless emperor have shown the most interesting results in terms of variability and presence of residual traces, especially in the area of the cuirass and on the robe.

On the statue, 11 points were analysed with XRF and 22 with FORS (Figure 3). All the measured spots and additional details were documented using a portable microscope. The divergence in the number of measured points depends on surface accessibility by the head of the two instruments.

The UVL images of the different pieces of the statue show several distinct luminescence values. Some of these areas can be related to integrations of modern materials, especially in the connection among the pieces and on the right leg, where some drips of modern material are visible. In the area of leather garment besides the materials with luminescence there are also visible areas with an absence of luminescence, probably corresponding to the marble integrations (Figure 4).

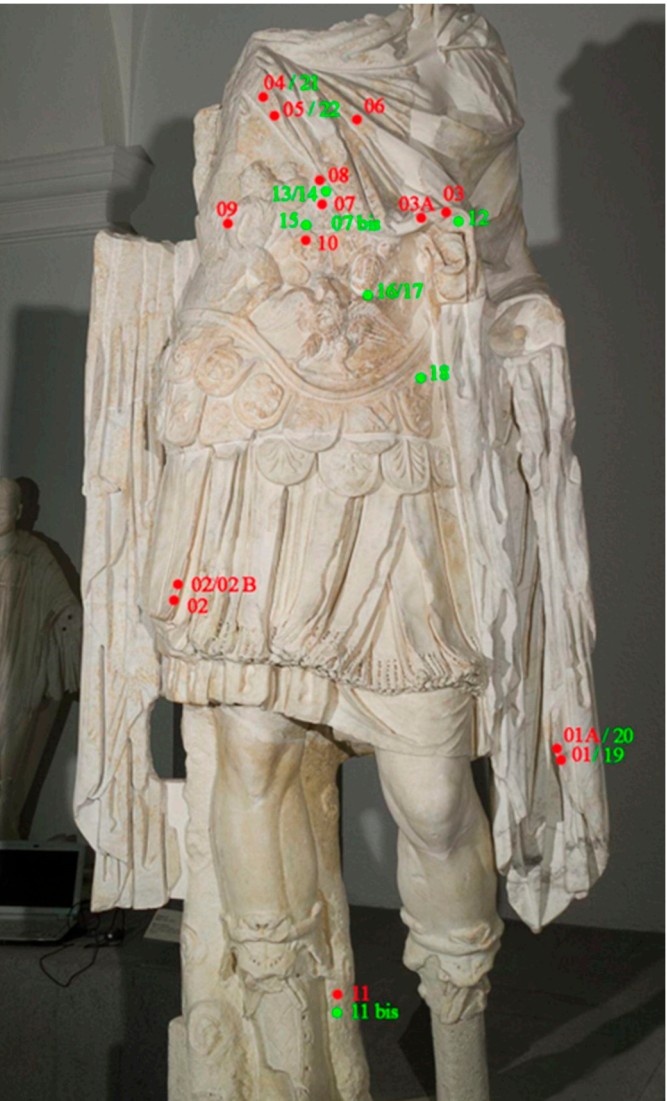

**Figure 3.** Cuirassed headless Emperor highlighting in red the spots analysed using both XRF and FORS; in green, extra points analysed using FORS.

The most evident aspect highlighted by the UVL is the different behaviour between the upper and lower part of the statue. While the upper part shows a very homogeneous and intense yellow luminescence, the lower one, circumscribed by the fractures, looks entirely cleaned and shows only a few areas with luminescence (Figure 5) [33,34].

These areas are more similar to those of the other statues, and they do not show the uniformity of the upper part.

The presence of this homogeneous luminescence opens some hypotheses for its explanation. The first is the presence of a ground layer still preserved; however, this luminescence disappears on the lateral side of the statue, in particular on the left and the interior area of the folder. While a consumption of the layer can be argued for the side, the same is not true for the inner part of the folder.

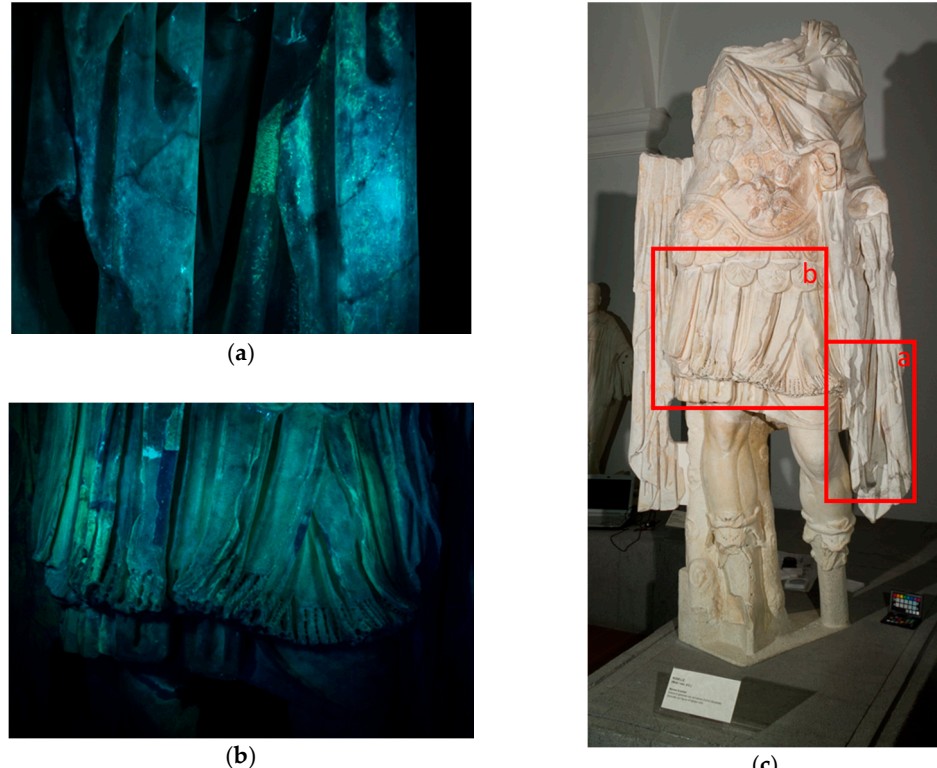

**Figure 4.** Different UV luminescence registered on the marble surface of the cuirassed headless Emperor statue (**a**,**b**) as highlighted (**c**).

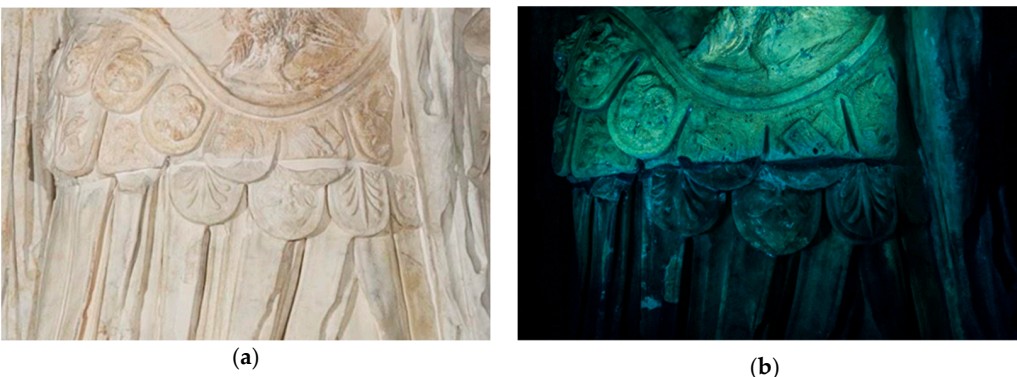

**Figure 5.** Detail of the visible image (**a**) and of the UV luminescence visible on the cuirassed (**b**).

The second is the presence of an organic material applied on the surface during a conservative treatment. This hypothesis finds support in the observation of the luminescence on the right side of the leather garment. Although the luminescence appears less intense, due to the presence of less substance, it is possible to observe the same luminescence in this area (Figure 6).

The pattern suggests the presence of an organic substance applied on the surface, not identified.

The observation of details on an optical portable microscope seems to support the hypothesis of a recently applied organic material.

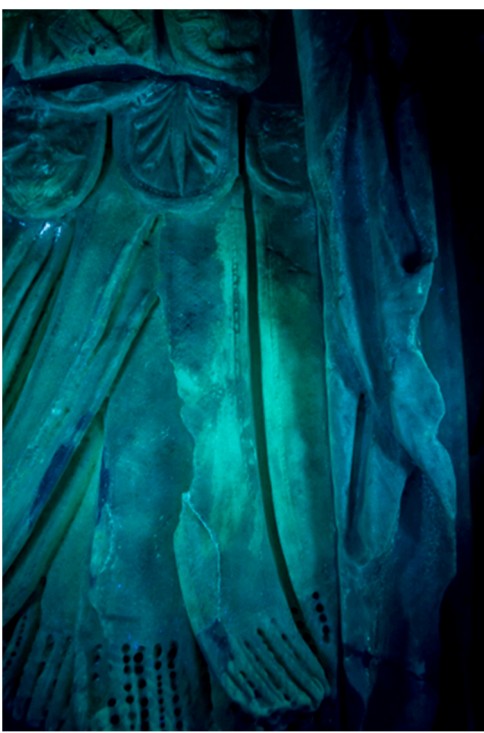

**Figure 6.** Detail of the *pteryges* of the cuirassed headless emperor statue, UVL.

In Figure 7a, the yellow glossy material goes over the red trace of pigment, while in the lower left corner in Figure 7b, the thickness of this material is clearly visible.

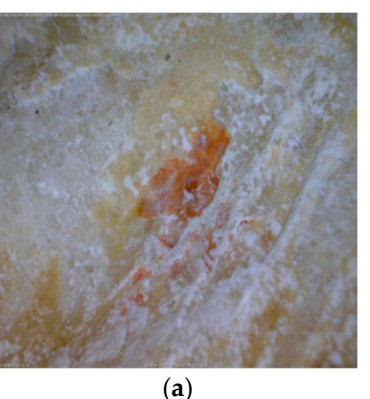 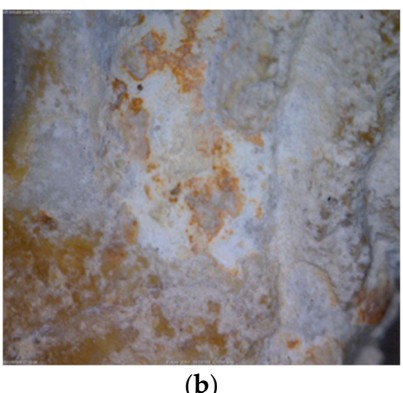

(**a**)               (**b**)

**Figure 7.** MD images of yellow glossy material on the surfaces of the cuirassed headless emperor statue (**a**,**b**).

Considering all the collected data, the second hypothesis of an organic material is the most consistent.

In a preliminary inspection of the statue's surface, it is possible to distinguish tiny traces of red, light brown, pale blue/grey and light yellow. Except for red traces, roughly visible on the whole statue, the other traces are focused on the cuirass where the relief surfaces helped to preserve the traces.

Evidence of a light yellow/brown layer is visible on the trophy's head and above it (Figure 8). The UVL images of the area show the presence of the previously mentioned bright luminescence, which tends to disguise the real luminescence of the traces. On the trophy's head, the high absorbance of ultraviolet radiation suggests the presence of an iron-based pigment [33].

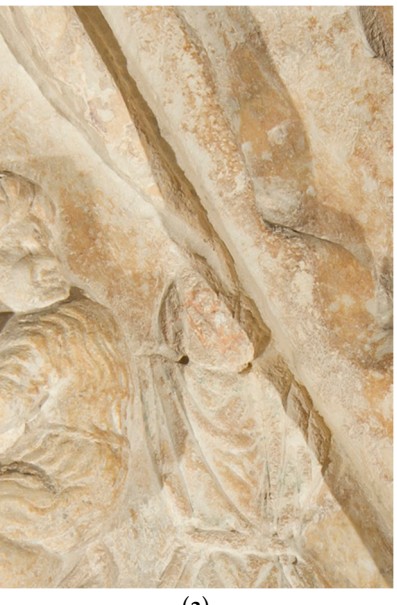
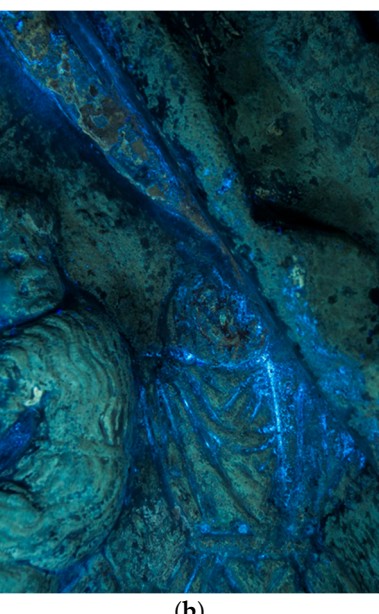

(**a**) (**b**)

**Figure 8.** Detail of the cuirass, visible light (**a**) and UVL (**b**) of the cuirassed headless emperor statue.

The traces on the folder, just above the head, slightly absorb the ultraviolet radiation, while the material on the background, visible on the surroundings, shows an intense yellowish luminescence. Comparing the luminescence of the pale pink areas of the head and upper folder, leaning to dark but with a slight yellow luminescence, with the residual background, it is evident that the luminescence is caused by a substance on top of both layers. This evidence confirms the hypothesis of a restoration material covering the surface.

Analyses performed using XRF on the trophy's head registered, besides the calcium, the presence of lead, iron, and zinc. The latter can be related to an impurity of the earth pigments, and it is well-documented in some Renaissance paintings and Hellenistic earth pigments varying in concentration, according to raw material provenance [35–37].

Red traces are visible in the lower part of the statue, on the hilt and on some plates of the leather garment. Through our analyses, it was possible to identify iron-based pigments, such as red ochres, by the presence of iron signals registered in the XRF spectra and by the presence of characteristic absorption bands in FORS spectra, due to iron oxides [31,32].

The VIL technique was used to investigate the whole surface, especially the breastplate, showing the presence of Egyptian blue mainly in two points of the cuirass: near the robe of the barbarian woman and on the trophy (Figure 9). Analyses on the area where blue was highlighted confirm the presence of Egyptian blue by the presence of signals of copper in XRF and through comparison with reference standards (Figure 9c,d) [31,32].

Scattered traces of Egyptian blue appear all along the cuirass of the emperor; however, they could be due to an unintentional mechanical diffusion of the pigment.

The analyses on the statue suggest a massive restoration work performed in different stages (as already reported). Despite this, the main traces of colours survived on the area of the *torso*, especially on the cuirass, where the folders of the decoration allow the protection of the layers of pale pink, blue and red pigments. A preliminary autoptic inspection revealed a wide light purple area in the lower part of the statue, in correspondence with the folds of the cloak (Figure 10).

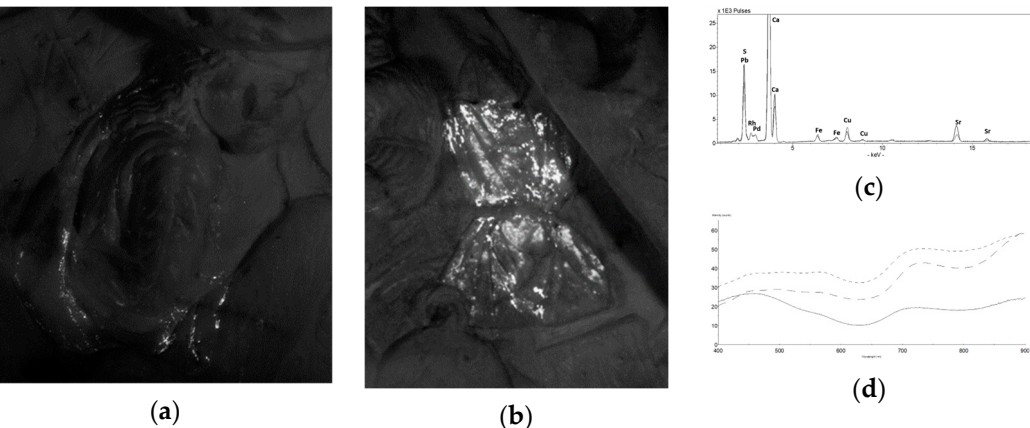

**Figure 9.** Details of the VIL of Egyptian blue on the cuirassed headless Emperor statue (**a**,**b**); XRF spectra of the two areas (**c**); FORS spectra of the two areas compared (dotted lines) with the spectrum of a reference of Egyptian blue (continuous line) (**d**).

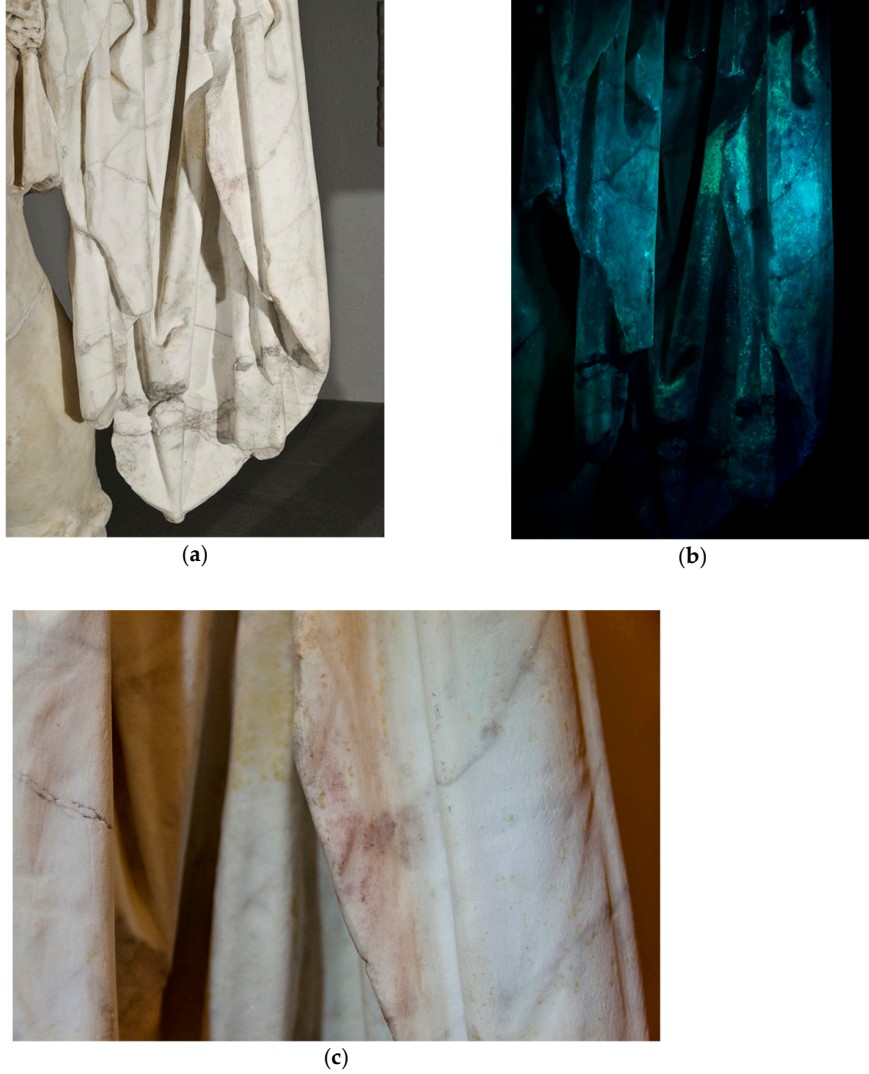

**Figure 10.** Detail of the mantle, visible light (**a**,**c**) and UVL (**b**) of the cuirassed headless emperor statue.

UVL documentation of the area with light purple/pink traces does not highlight any characteristic luminescence, thus pointing out the absence of any organic colourant, such as an organic red lake. XRF shows the presence of traces of gold and iron, while the FORS spectrum is compatible with an iron-based pigment.

High magnification images documented the presence of small particles of gold widespread on the surface, suggesting the presence of a gilded decoration (Figure 11).

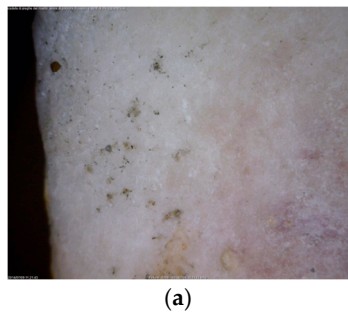 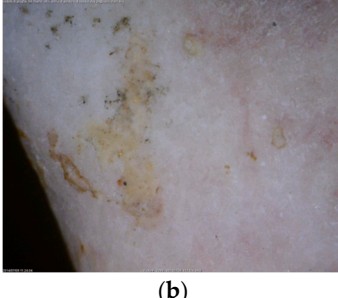

(**a**) (**b**)

**Figure 11.** High magnification images, detail of the mantle, small traces of gold on the cuirassed headless emperor statue (**a**,**b**).

In addition to that, small amounts of gold were found all over the edge of the mantle, corresponding to measure points n.3, 3A and n.5, and on the surface of one of the edges of the leather garment (n.2 and 2A) (see Figure 2). In many cases, these traces were only slightly detectable through a visual inspection and the use of a portable microscope was of great help, providing a more detailed vision of these areas.

The most consistent gold traces, associated with purple/pink areas, can be observed on a fold in the upper part of the statue, in correspondence with the mantle (n.4). In this area, the residual polychromy is well-visible, and the hue is darker, with a more purple shade as highlighted using the microscope (Figure 11).

Traces of gold were also found on leather garments always associated with a purple shade (Figures 12 and 13).

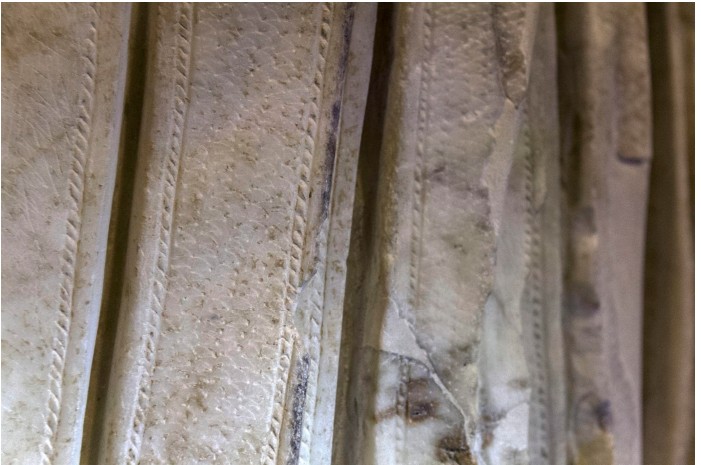

**Figure 12.** Visible light, detail of the *pteryges* with purple hues on the cuirassed headless emperor statue.

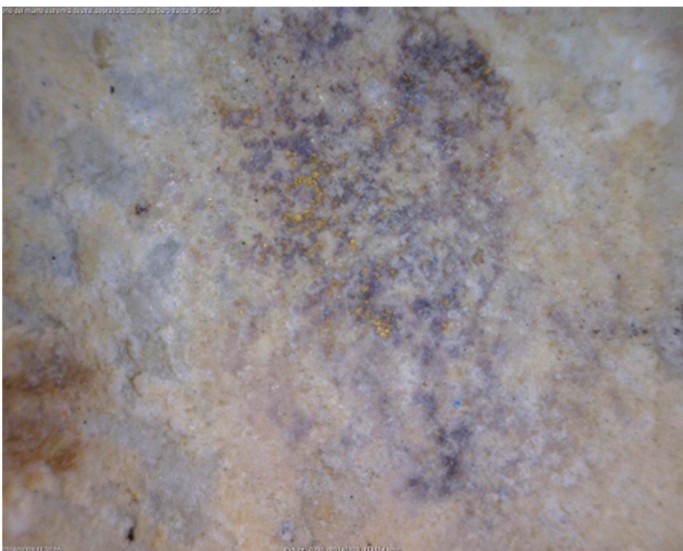

**Figure 13.** High magnification image, purple shade and gold traces on the cuirassed headless emperor statue.

In addition to high magnification images, in this area, macro-photography contributed to observe and document the pattern used to reproduce the stitching decoration of the leather garment.

These preliminary non-invasive analyses cannot infer any hypothesis on the coexistence of gold and these pink/violet traces. Hypotheses based on similar traces found on other published analyses suggest the presence of a bolus-like substance to provide a uniform layer for the adhesion of the gold on the surface or the results of a gold degradation into a colloid material [38,39].

The literature provides a distinction among the aspects of colloidal gold in shape and dimension, suggesting a red hue for particles measuring less than 100 nm in size and with a round shape, while a blue/purple shade appears with elongated particles with dimensions higher than 100 nm [40].

In addition to the shape and dimension of the particles, the presence of organic deposits, silica, local refractive index, and other materials can also alter the colour of colloidal gold [41].

Since both colloidal solutions cannot be discriminated through a sole observation using an optical microscope, two samples were acquired for SEM-EDS analysis.

They were observed and documented with an optical microscope and later with SEM, showing the presence of pink shade on both fragments. Despite this, only one fragment, called fr_2, shows a visible flake of gold (Figure 14).

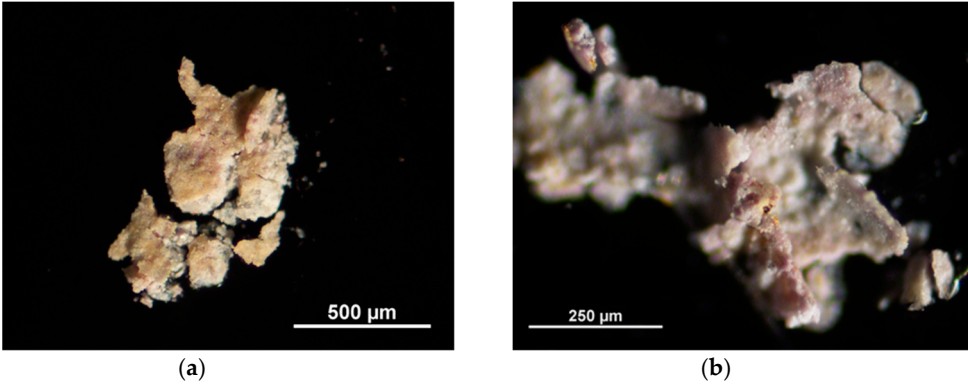

| (a) | (b) |

**Figure 14.** Sample no.1 (fr_1) at 4× magnification (**a**). Sample no.2 (fr_2) at 10× magnification (**b**).

The gold portion appears as a preserved chip with a dimension of ~26 μm in a backscattering image. Another two chips are visible in the area; however, they are longer than 7 μm (Figure 15).

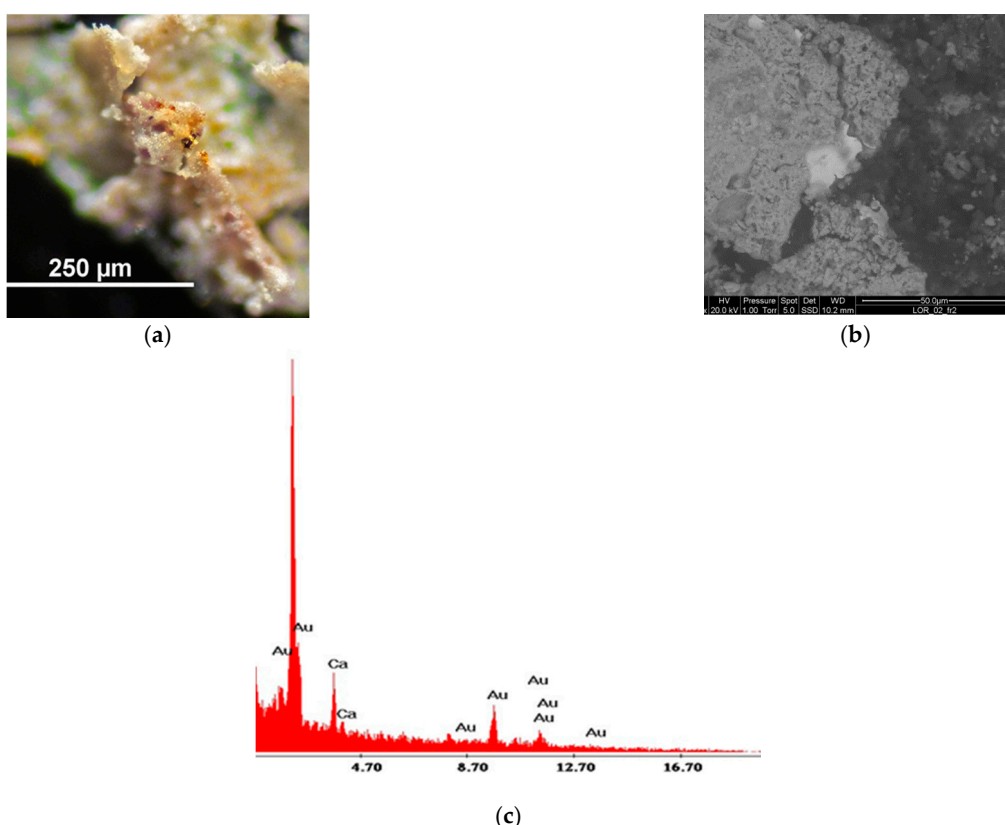

(a)

(b)

(c)

**Figure 15.** Image at 20× magnification (**a**). The backscattered image of the area (**b**) and EDS microanalysis of gold chip (**c**).

SEM analysis of fragments in the areas corresponding to the pink hue, in both cases, do not show any presence of gold nano-particles. Hence, it is possible to exclude the colloidal gold for the presence of these pink/purple areas.

As already discussed, the absence of luminescence, excluding the presence of organic lakes. However, at the moment and with the protocol used, it is not possible to speculate on the origin of these colours.

## 4. Discussion

*4.1. The Cuirassed Statue: The Breastplate*

From the typological point of view, the cuirassed statue is very characteristic: the mantel is falling from the right shoulder across the breast and covering the sword on the left side, and the right arm is risen, probably holding a spear. The comparison with a little group of similar statues [42] (pp. 324–326) allows the researchers to date them between the Flavian and the early Trajanic age.

The decoration of the breastplate is unparalleled: at the foot of the trophy is a female prisoner, while a northern barbarian is arriving from the left, holding a child [43] (p. 166). We can interpret this scene as the offering of a hostage, according to a scheme known in the 30th scene of the Trajan Column [44] (pp. 221–222, tav. 37), where a mother is offering her child as a hostage to Trajan, or on the fragment of a cuirassed statue of the Ny Carlsberg Glyptotek [45] (Figure 1, pl. 49); [46] (pp. 62–63, nr. 31, pl. 46); [43] (pp. 84, 166, Figure 82). Moreover, Tacitus and Cassius Dio mention offerings of hostages, respectively, under Claudius and Domitian by the German tribes of the Chatti and Cherusci (Claudius: Tac., *Ann*. XII.28; Domitian: Dio 67.10.5). All these elements suggest an allusion to the

triumph on the Chatti in A.D. 82–83 and lead to the portrait being identified as Domitian, who, on that occasion, assumed the title of Germanicus. The proposed identification with Claudius [19] is untenable for typological reasons and the connection with the speech to Claudius of Caratacus king of the Catuvellauni (Tac., Ann XII.36,37) is based on excessively generic evidence. Further elements reinforce the likelihood of the identification with Domitian: in the Flavian age, the *forum* of *Rusellae* reached his final appearance due to the paving of the square with stone slabs and, in the same period, two members of the gens Viciria from *Rusellae* entered the senate in Rome, obtaining two consuls in 89 (A. Vicirius Proculus) and 98 A.D. (A. Vicirius Martialis). In the following centuries, this favourable situation was not to be repeated.

No traces of colour survive on the skin or on the ground of the cuirass. The evidence of the other sculptures, where red ochre was found, suggests that the whole cycle was completely painted and, in general, togate statues or female-draped statues do not offer insurmountable obstacles for a hypothetical reconstruction. More difficult is the problem of the cuirassed statues, about which we do not have—to date—any sure indication concerning the colour of the ground of the breastplate.

Searching for comparisons, we could find some evidence from the Alexander Sarcophagus in Istanbul [47], from the frescoes of the Macedonian tombs [48] or—in Italy—from the Amazon Sarcophagus in Florence [49], the Samnite tomb of Nola [50], or the Alexander Mosaic from Pompeii [51]; however, all of these examples are quite remote, both from the chronological and the typological point of view. In a later period, we have a 4th century fresco from the late antique domus of *Via dei Laterani* (Rome) with what is thought to be the figure of Mars wearing a cuirass; however, the surface is badly damaged and not very helpful [52] (Figure 112), [53] (p. 148).

Leaving linen and leather aside, the ancient sources mention imperial breastplate in bronze, iron, silver, or gold. If we consider that bronze and iron are usually painted blue, a colour completely lacking on the ground but well-attested for on the relief, and that gold is attested for with traces of gilding on the hem of the mantle, the easiest solution is to hypothesize a white ground portraying a silver breastplate. The same hypothesis was also considered for the interpretation of the white background for the polychrome neoattic krater from Oplontis, Villa A [54]. We remind the reader that the same solution was also recently hypothesized for the cuirass of the Prima Porta Augustus [20]. In that case, the authors raised an issue about the function of colours in a cuirassed sculpture: are they colours of a metallic breastplate, the colours used in ceremonial armours? Or are they indeed the colours that a painter would use to depict—on marble painting or fresco—a subject such as that? The first hypothesis could lead to a choice of realism of the details in the metal elements, thus leading beyond the painting on marble, in the field of metallurgy and the possibilities concerning the tincture of metals.

The second hypothesis, in contrast, leads to an unrealistic use of the colours whose reference is only to the scene sculptured on the cuirass: in this case, a trophy and three barbarian people—a woman, and a man bringing a child towards the trophy.

### 4.2. The Cuirassed Statue: The Hemline

If we consider the hemline of the mantle, we can compare it with other examples of mantles with painted border along the hem, such as the mantle of Theseus in the painting on a marble slab from Herculaneum [55] (p. 105), or, more similar to the gilded hem of the cuirassed sculpture from *Rusellae,* the yellow hem of Trajan's portrait from Samos [56,57].

### 4.3. Colour for Backgrounds

The last point is the red colour on the base of the little part of a second cuirassed statue. There is no trace of colour on figural elements; however, there are evident traces of red on the rough ground of the marble between two elements: a helmet and a trunk. This was part of the painted decoration for a non-figurative element, which had its own colour: in this case, red. As we know, the red colour—red ochre in particular—could only be a

preparation for another layer of a different pigment. Therefore, we cannot hypothesize the original colour of this element; however, we can consider it another example of coloured background from the Imperial age [58–60].

## 5. Conclusions

The whole imperial cycle of sculptures from *Rusellae* was coloured, as can be observed from the traces of pigments still visible on the statues. Moreover, the upper part of the cuirassed statue shows a significant amount of traces of its original colours, such as Egyptian blue for the mantle of the barbarian woman and the vest of the trophy (in the last case blue is mixed with lead white), red ochre for the head of the trophy, and gold for the hem of the cloak of the emperor. The ground of the cuirass was probably white, thus representing a silver breastplate. All this information provides some new data about the painting on imperial portraits, and especially on cuirassed portraits between the end of the first century B.C. and the end of the first century A.D.

**Supplementary Materials:** The following supporting information can be downloaded at: https://www.mdpi.com/article/10.3390/heritage6040179/s1, table with complete description of each statue.

**Author Contributions:** Conceptualization, P.L., S.B., R.I., S.L. and D.M.; methodology, P.L., S.B., R.I., S.L. and D.M.; software, S.B., R.I. and D.M.; formal analysis, S.B., R.I. and D.M.; investigation, S.B., R.I. and D.M.; resources, P.L. and S.L.; data curation, S.B., R.I, and D.M.; writing—original draft preparation, P.L., S.B., R.I., S.L. and D.M.; writing—review and editing, P.L., S.B., R.I., S.L. and D.M.; visualization, P.L., S.B., R.I., S.L. and D.M.; supervision, P.L., S.B., R.I., S.L. and D.M.; project administration, P.L., S.B., R.I., S.L. and D.M. All authors have read and agreed to the published version of the manuscript.

**Funding:** This research received no external funding.

**Data Availability Statement:** The data presented in this study are available in the article and in the Supplementary Material.

**Acknowledgments:** The authors wish to thank Gabriella Poggesi of the former Soprintendenza per i Beni Archeologici della Toscana; the former Director of the Maremma Archaeology and Art Museum in Grosseto, Mariagrazia Celuzza, the restorer Cristina Barsotti, and all the staff of the Museum.

**Conflicts of Interest:** The authors declare no conflict of interest.

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
