# Peer review of "New Evidence about the Polychromy of Early Imperial Cycle from the Augusteum of Rusellae (Tuscany)"

_heritage, doi:10.3390/heritage6040179_

Round 1
Reviewer 1 Report
The article “New evidence about the polychromy of Early Imperial cycle 2 from the Augusteum of Rusellae (Tuscany)” reveals interesting information concerning the presence of polychromic and gilding traces on marble statues. The adopted methodology is not innovative; however, the obtained results are useful to contribute on pictorial technique adopted in the imperial cycle of sculptures from Rusellae. I recommend publication after minor revision. See comments below.
Line 23: it should be better to explain which type of information you obtained.
Line 77: here you state that the traces of colors were preliminarily documented with a portable microscope; is this microscope the same one used to document the areas analyzed with XRF and FORS? Please specify this point in the M&M section.
Line 96: it would be useful to provide an image of all the analyzed artworks in the M&M section or in a supplementary file. Here, you could also underline in smaller images, the details of the colored areas
Line 112, Table 1: please provide a better alignment of the “Filters on objective column”.
Line 133: mm2, make the 2 subscripts.
Line 187: please specify in Figure 3 which statue you are analyzing, and I suggest using (a), (b), and (c) in the figure and in the caption to specify the adopted lights.
Line 194: figure 4, I suggest using (a) and (b) in the figure and in the caption to specify the adopted lights.
Line 213: please specify in Figure 5 which statue you are analyzing.
Line 220: please specify in Figure 6 which statue you are analyzing.
Line 228: figure 7, I suggest using (a) and (b) in the figure and in the caption, rather than left and right.
Line 254: please remove “-“ from the caption.
Line 264: please specify in Figure 9 which statue you are analyzing, and I suggest using (a), (b), and (c) in the figure and in the caption to specify the adopted lights.
Line 274: please specify in Figure 10 which statue you are analyzing, and I suggest using (a) and (b) in the figure to give more details about the images.
Line 286: please specify in Figure 11 which statue you are analyzing,
Line 288: please specify in Figure 12 which statue you are analyzing,
Line 313: I suggest using (a) and (b) in figure 13, rather than left and right.
Lines 316, and 318: please remove the bold style from μ.
Line 325: I suggest using (a), (b), and (c) in figure 14, both in the figure and in the caption rather than left, center, and right.
At the end of the results paragraph, it should be better to insert a final table with XRF and FORS results obtained from the different analyzed statues and points.

Reviewer 2 Report
The manuscript is clear, relevant, and well-structured but has some errors that the authors must improve for publishing. The references they used are relevant, but they must add more references to justify some affirmations and support the article's scientific soundness.
Generalities
- Please cite all the figures in the text.
- In the abstract put, "two micro samples from the headless cuirassed statue were also analyzed by ART-FTIR and EDS-SEM" (lines 21-22), but in the text, you did not mention anything about ART-FTIR. Please, add the results and the interpretation of the analyses with this technique.
- Add more modern and actual references and/or literature to support some affirmations.
- In the References, please, add the DOI of the publications.
Abstract
Line 13: Please, add the country after Tuscany.
1. Introduction
It could be interesting to add a map with the site's location, an archaeological plan, and photos of the Augusteum.
Line 44: Please, use italics for "portico". The words in another language should be written in italics.
1.2. Polychromy
Please, talk about the colors that you can see in the sculptures and describe them.
2. Materials and Methods
Line 133: Please, use superscript in mm2.
Please, add sampling criteria, where the samples were taken from, in which areas, how the samples were taken, the preparation of the samples carried out for each analysis technique, etc.
Add software used with each technique.
3. Results
Add a table with the summary of every statue studied, the number of samples, the color of the sample, and the technique used for the analysis.
There are no documents in the museum that talks about the interventions taking place in these sculptures in the past? Did you check it? It could be fascinating to confirm the modern materials founds in the statues studied. Please, add a paragraph explaining these.
Line 158: Please, add references before "interventions" to support this affirmation.
Line 162: Please, add references to support this affirmation.
Line 182: Please, explain better the characterization of organic materials using UVL images with the support of other works and references that made it.
Line 186: Please, add references to support this affirmation.
Lines 198-207: Please, add references to support these hypotheses.
Line 208: Something? Please, specify more or describe it better.
Line 225. Please, can you provide a photograph with all of these colors?
Line 234: Please, add references to support this affirmation.
Line 240: Please, add references to support this affirmation.
Please, add more information, discussion, and references about how the pigments were identified with FORS and XRF. Add some reflectance spectrum obtained with FORS and the spectrum and percentages obtained with XRF.
Line 248: Please, add references to support this affirmation.
Line 252: How do you identify Egyptian blue? Please, explain and add references to support this.
Line 257: Please, add references to support this affirmation.
Line 260: Please, use italics for "torso".
Lines 267-270: Please, add spectrum and references in this paragraph.
Line 277: This numeration of the samples needs to be explained in the text.
Line 316 and 318: Quit bold font.
Figure 14: Please, explain all the elements present in the spectrum. Change the EDS microanalysis image; an axis is missing.
4. Discussion
Line 365: Please use italics with the words of other languages.
Line 372: Please, add references.
Line 381: Please, add references.
5. Conclusions
The scientific soundness of the article is better with an impersonal subject.
Lines 404 – 406: Explain better how you can specify the type of pigments and their composition without elemental and compositional analysis. Add references.
Lines 411-423: This is not a conclusion for the manuscript presented; please, move it to the introduction when you talk about the Augustem.
